# Effect of Different Titanium Dental Implant Surfaces on Human Adipose Mesenchymal Stem Cell Behavior. An In Vitro Comparative Study

Vittoria D'Esposito [1], Josè Camilla Sammartino [2], Pietro Formisano [3], Alessia Parascandolo [3], Domenico Liguoro [1], Daniela Adamo [4], Gilberto Sammartino [4,*] and Gaetano Marenzi [4]

[1] URT "Genomics of Diabetes", Institute of Experimental Endocrinology and Oncology, National Research Council, 80131 Naples, Italy; vittoria.desposito@unina.it (V.D.); domenico.liguoro@cnr.it (D.L.)

[2] Department of Biology and Biotechnology "L. Spallanzani", University of Pavia, Via Ferrata 1, 27100 Pavia, Italy; jose.sammartino@iusspavia.it

[3] Department of Translational Medicine, University of Naples "Federico II", 80138 Naples, Italy; fpietro@unina.it (P.F.); parascandolo@ceinge.unina.it (A.P.)

[4] Department of Neurosciences, Reproductive and Odontostomatological Sciences, University of Naples "Federico II", 80138 Naples, Italy; daniela.adamo@unina.it (D.A.); gaetano.marenzi@unina.it (G.M.)

[*] Correspondence: gilberto.sammartino@unina.it; Tel.: +39-(817)-462-118

**Abstract:** Background: The aim of this research was to evaluate the effects of three different titanium (Ti) implant surfaces on the viability and secretory functions of mesenchymal stem cells isolated from a Bichat fat pad (BFP-MSCs). Methods: Four different Ti disks were used as substrate: (I) D1: smooth Ti, as control; (II) D2: chemically etched, resembling the Kontact S surface; (III) D3: sandblasted, resembling the Kontact surface; (IV) D4: blasted/etched, resembling the Kontact N surface. BFP-MSCs were plated on Ti disks for 72 h. Cell viability, adhesion on disks and release of a panel of cytokines, chemokines and growth factor were evaluated. Results: BFP-MSCs plated in wells with Ti surface showed a viability rate (~90%) and proliferative rate comparable to cells plated without disks and to cells plated on D1 disks. D2 and D4 showed the highest adhesive ability. All the Ti surfaces did not interfere with the release of cytokines, chemokines and growth factors by BFP-MSCs. However, BFP-MSCs cultured on D4 surface released a significantly higher amount of Granulocyte Colony-Stimulating Factor (G-CSF) compared either to cells plated without disks and to cells plated on D1 and D2. Conclusions: The implant surfaces examined do not impair the BFP-MSCs cell viability and preserve their secretion of cytokines and chemokines. Further in vitro and in vivo studies are necessary to define the implant surface parameters able to assure the chemokines' optimal release for a real improvement of dental implant osseointegration.

**Keywords:** dental implant surface; chemokines; grow factors; bone damage; osseointegration

## 1. Introduction

Successful implant osseointegration requires maximal bone–implant interaction for promoting de novo bone formation on the surface through a continued recruitment and migration of differentiating osteogenic cells to the implant site during the contact osteogenesis. An involvement of the immune system during the osseointegration process was reported by Trindade et al.; the activation of type 2 macrophages (M2-macrophages) tended to facilitate the osteogenic behavior of mesenchymal stem cell (MSCs) and attract fewer inflammatory cells improving the clinical performance of the dental fixtures manipulating the balance of bone regeneration/absorption [1]. The increase in macrophages secretion of macrofage colony-stimulating factor (M-CSF), IL-1ß, IL-6, osteoprotegerina (OPG) and transforming factor beta (TGF-ß) indicated that the activation of the osteoblast can be induced by immunological factors secreted by macrophages and MSCs. Ma et al. reported the effects of some surface parameters on the inflammatory response of the macrophage

immune cells and bone MSCs [2]. The author suggested that understanding and monitoring the profile of cytokines secreted by macrophages and the retroregulative cytokines released by MSCs can provide a framework for systemically analyzing and predicting the performance of a fixture [2]. Many studies reported the improvement of the bone–implant interface and the titanium bioactivity by modifying the surface roughness, hydrophilicity and wettability [3–10]. Implant surface treatments (acid-etching, grit-blasting, a combination thereof, Ti plasma spraying) were suggested in order to obtain a rough surface increasing the contact with the peri-implant bone [11,12]. However, the ideal value of the micro-roughened profile is still controversial, many investigations reported a real improvement of the osseointegration with an arithmetic mean deviation of the roughness profile (Ra) ranging between 1.0 and 2.0 μm [13–17]. The Ti surface modification promoted the enhancement of regenerative cell adhesion that leads to reduction of the ability of bacteria to form biofilms and, as consequence, triggers infection [18,19]. This requires the recruitment of MSCs able to adhere to the implant surface, the consequent chemokines and growth factors release and their osteoblast differentiation [18,20]. Adipose-derived stem cells taken from Bichat's buccal fat (BFP-MSCs) demonstrated their osteogenic differentiation when used in maxillofacial surgery with a regenerative aim [21]. Since they demonstrated a substantial capacity for bone formation, they are an excellent model to evaluate cell–implant connection [15,19]. Whether BFP-MSCs preserve their viability and secretory functions in the presence of Ti disks is unknown. Thus, the aim of the current work was to evaluate the effect of three different Ti dental implant surfaces on the viability and release of multiple cytokines, chemokines and growth factors by human mesenchymal stem cells obtained from a Bichat fat pad. This is a novel approach since it will provide new knowledge regarding the interaction between Ti implants and local or injected BFP-MSCs.

## 2. Materials and Methods

### 2.1. Preparation of Specimens

In this study, four different Ti disks were used as substrate: (I) smooth Ti as control (D1); (II) pure Ti grade 4 (chemically etched), which reproduced the Kontact S surfaces (D2); (III) Ti grade 5 (Sandblasted), resembling the Kontact surfaces (D3); (IV) pure Ti grade 4 (blasted/etched), which reproduced the Kontact N surface (D4).

As reported by the manufacturer, the acid-etching of D2 surface was performed with hydrofluoric acid (HF); the grit-blasting of D3 was performed with alumina ($Al_2O_3$); in the D4 surface these techniques were used in combination to homogenize the microprofile of the surface and to remove as much as possible the residual blasting particles. The rugosimetric survey of the dental implant surfaces resulting from the different treatments was previously reported; it was carried out through a Leica Definition Confocal Microscopy (DCM) 3 D (Leica Microsystems, Schweiz, AG-CH, Heerbrugg, Switzerland) equipped with LeicaScan and LeicaMap software (Leica Microsystems, Schweiz, AG-CH, Heerbrugg, Switzerland). The reported amplitude parameters were as follows: Ra as the arithmetical mean of the sum of roughness profile values; Rp as the maximum peak height of the respective profile; Rv as the maximum peak height of the respective profile; Rz as the maximum height of the respective profile. The average value of the amplitude parameters (Ra, Rp, Rv and Rz) of the implant surfaces are reported in Table 1 [15].

**Table 1.** Roughness measurement of the dental implants estimated through the amplitude parameters Ra, Rp, Rv, Rz taken with Leica Map.

| Amplitude Parameters | Kontact | Kontact S | Kontact N |
|:---:|:---:|:---:|:---:|
| Ra | 0.86 | 1.28 | 1.4 |
| Rp | 2.27 | 2.66 | 4.8 |
| Rv | 2.82 | 4.82 | 3.00 |
| Rz | 5.09 | 7.48 | 7.8 |

Disks had a 12 mm diameter and a 1 mm thickness and were provided by Biotech Dental (Salon de Provence, France).

The surface of control ant test disks were cleaned with purified water, enzymatic detergent, acetone and alcohol.

### 2.2. Bichat's Fat Pad (BFP) Tissue Collection and Cell Isolation

BFP-MSCc samples were collected after maxillary bone reconstruction interventions [21]. Once obtained, BFP samples were quickly dispatched to the laboratory for cell isolation. Adipose tissues were reduced to small pieces and subsequently subjected to enzymatic digestion using collagenase solution (1 mg/mL, from Sigma-Aldrich, St. Louis, MO, USA) at 37 °C for 1 h. BFP-MSCs were isolated from the stromal vascular fraction through centrifugation ($1200\times g$ for 5 min) and plated in Dulbecco's modified Eagle's medium (DMEM)-F12 (1:1) supplemented with 10% fetal bovine serum, 2 mM glutamine, 100 U/mL penicillin and 100 U/mL streptomycin (Lonza Group Ltd., Basel, Switzerland). Isolated BFP-MSCs showed an ability to differentiate in adipocytes evaluated by lipid accumulation staining (Oil Red O) and in osteoblast-like cells evaluated by Alizarin Red staining. Moreover, the expression of mesenchymal markers CD-73 and CD-90 was confirmed by flow cytometry, as we previously reported [21].

Investigations were carried out following the rules of the Declaration of Helsinki of 1975, revised in 2013. Informed consent was obtained from every subject before the procedure. Protocol was approved by the ethical committee of the University of Naples "Federico II" (prot. n. 89/15).

### 2.3. Cell Viability Assay

BFP-MSCs were seeded on to different Ti surfaces and assessed for cell proliferation and viability using Trypan blue staining as previously described [22]. In detail, $3 \times 10^4$ BFP-MSCs were plated on Ti disks in 12-well culture plates. Ti disks and cells were completely immersed in standard culture medium. After 72 h, culture media were soaked, disks were moved to a new 12-well culture plate and cells (termed cells "on disks") were collected by trypsinization and counted using a 0.4% Trypan Blu dye exclusion test (Sigma-Aldrich), used to determine the number of viable cells present in a cell suspension [23]. Cells remained in the original culture plate (termed cells "under the disks") were also counted. As control (CTRL), BFP-MSCs were plated without disks, in standard growth conditions. Briefly, 10 μL of Trypan Blu staining solution were added to 10 μL of cell suspension, and 10 μL of the final mix were inserted in cell counting slides. Cells were visualized and counted by using the TC10$^{TM}$ automated cell counter (Bio-Rad, Hercules, CA, USA). Based on Trypan Blu staining, the cell counter provided information about total cell number, live cell number and percentage of viable cells. Subsequently, cells harvested from the disks were plated in 12-well culture plates without disks, photographed, and counted after 72 h. Images were taken by the Olympus DP20 microscope digital camera system (Olympus Corporation, Tokyo, Japan) associated to a phase contrast inverted microscopy (Olympus Corporation, Tokyo, Japan) with 100× magnification

### 2.4. Bioplex Analysis

Conditioned media from BFP-MSCs plated on Ti disks for 72h were screened for the concentration of IL-1ra, IL-1b, IL-2, IL-4, IL-5, IL-6, IL-7, IL-8, IL-9, IL-10, IL-12(p70), IL-13, IL-15, IL-17, basic FGF, Eotaxin, G-CSF, GM-CSF, IFN-γ, IP-10, MCP-1, MIP-1α, MIP-1β, CCL5/RANTES, TNF-α, PDGF-BB and VEGF using the Bioplex multiplex Human Cytokine, Chemokine and Growth factor kit (Bio-Rad, Hercules, CA, USA) according to the manufacturer's protocol. The magnetic Bead–Based Assay was performed on a Bio-Plex 200 System (Bio-Rad, Hercules, CA, USA) [24].

*2.5. Statistical Analysis*

Data were analyzed with GraphPad Prism 7.0 software (GraphPad Software Inc., La Jolla, CA, USA). Results were presented as mean ± standard deviation (SD) of three independent experiments. Each experiment was performed with one Ti disk per group. For comparisons between 2 groups, a two-tailed *t*-test for independent samples was used. Multiple comparisons among more than two groups were made using the analysis of variance (ANOVA) test with Tukey correction. The *p* value of <0.05 were considered statistically significant.

## 3. Results

*3.1. Titanium Disks Do Not Impair BFP-Mesenchymal Stem Cell (MSC) Viability*

To evaluate the growth and viability of BFP-MSCs on Ti implants, cells were plated on Ti disks for 72h. Next, disks were moved to new culture plates and cells (termed cells "on disks") were collected by trypsinization and counted using 0.4% Trypan Blu dye exclusion test, used to determine the number of viable cells present in a cell suspension [23]. Cells remaining in the original culture plate (termed cells "under the disks") were also counted. As control (CTRL), BFP-MSCs were plated without disks, in standard growth conditions. Cells were detected on all Ti disks. BFP-MSCs on treated Ti disks showed a proliferative rate comparable to control and comparable to cells on untreated disks (D1) (Figure 1). Furthermore, when plated on Kontact S (D2) and Kontack N (D4) surfaces, the number of living cells counted on disks was significantly higher compared to that of cells under the disks (Figure 1). By contrast, on Kontact (D3) surface the number of living cells counted on the disk was similar to that of the cells under the disk.

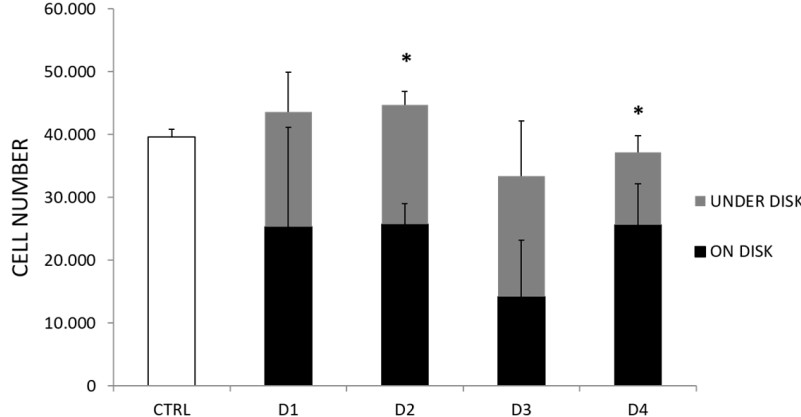

**Figure 1.** Bichat's fat pad mesenchymal stem cell (BFP-MSC) growth on Ti disks. Number of BFP-MSCs plated in 12-well culture plates without disks (CTRL), on D1, D2, D3 or D4 for 72 h. Black bars represent live cells counted on disks. Gray bars represent live cells counted under the disks. * Denote statistical significance of cells on disk versus cells under disk (* *p* < 0.05).

Moreover, Ti disks were not responsible of any cytotoxic effect on BFP-MSCs. Indeed, cells on disks displayed a 90% viability rate with no differences among disks and compared to control cells (Figure 2).

To investigate the long-term effect of Ti disks on BFP-MSC viability, cells detached from disks were re-plated. After 72 h, BFP-MSCs were live, attached on the plate, reflective, and with a morphology comparable to control cells (Figures 3 and 4). Moreover, they showed a proliferative rate of about three-fold, with no differences among the experimental conditions (Figure 4). Therefore, also after the contact with disks, cells retain their ability to survive and proliferate.

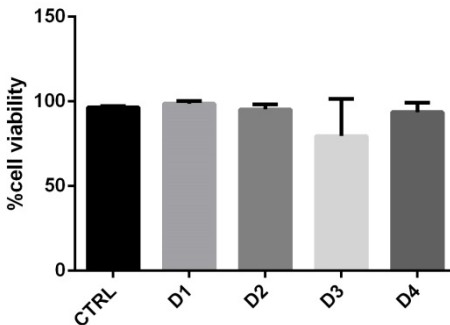

**Figure 2.** BFP-MSC viability on Ti disks. Percentage of live BFP-MSCs plated in 12-well culture plates without disks (CTRL), on disk D1, disk D2, disk D3 or D4 after 72 h.

CTRL

D1

D2

D3

D4

**Figure 3.** BFP-MSC morphology. Representative image of BFP-MSCs harvested from disks and re-plated. Images were taken by the Olympus DP20 microscope digital camera associated to a phase contrast inverted microscopy (magnification 100×).

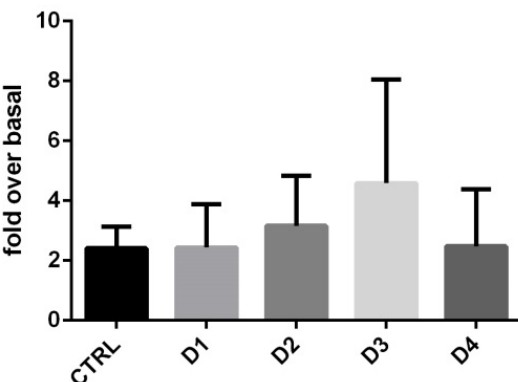

**Figure 4.** Number of BFP-MSCs harvested from disks and re-plated in 12-well culture plates for 72 h. Bars represent the number of live cells relative to those plated upon detachment from disks.

### 3.2. BFP-MSC Interaction with Ti Disks Do Not Alter Cell Secretion

It was evaluated whether Ti disks could interfere with BFP-MSCs' secretory profile. To this aim, we analyzed the release of a panel of cytokines, chemokines and growth factors in the medium of BFP-MSCs plated without disks and of BFP- MSCs cultured on D1, D2, D3, and D4. Among all the factors analyzed, IL-1ra, IL-2, IL-5, IL-6, IL-8, IL-9, IL-10, IL-12 (p70), IL-15, IL-17, basic FGF, G-CSF, IFN-$\gamma$, IP-10, MCP-1, MIP-1$\beta$, CCL5/RANTES, TNF-$\alpha$, PDGF-BB and VEGF were the factors more abundant in BFP-MSC medium (Figure 5). Overall, Ti surfaces did not interfere with cell secretory functions. However, when the cells were plated on D2 (chemically etched surface), a significant reduction of IL-1ra, IL-2, IL-10, IL-12, IL-17, and G-CSF, and a reduced trend of basic FGF ($p = 0.054$) was detected, compared to D3 (sandblasted surface). Interestingly, BFP-MSCs, in presence of D4 (blasted/etched surface), displayed a slight increase of IL-1ra ($p = 0.09$)-compared to D2- and a significant increase of G-CSF, compared to cells plated without a disk and to cells plated on D1 and D2 (Figure 5).

Taken together, all our data demonstrated that BFP-MSCs plated on different Ti disks are able to keep their own viability, adhesion and secretory capacity. Cells plated on D4 showed high adhesion on the disk and released a higher amount of G-CSF.

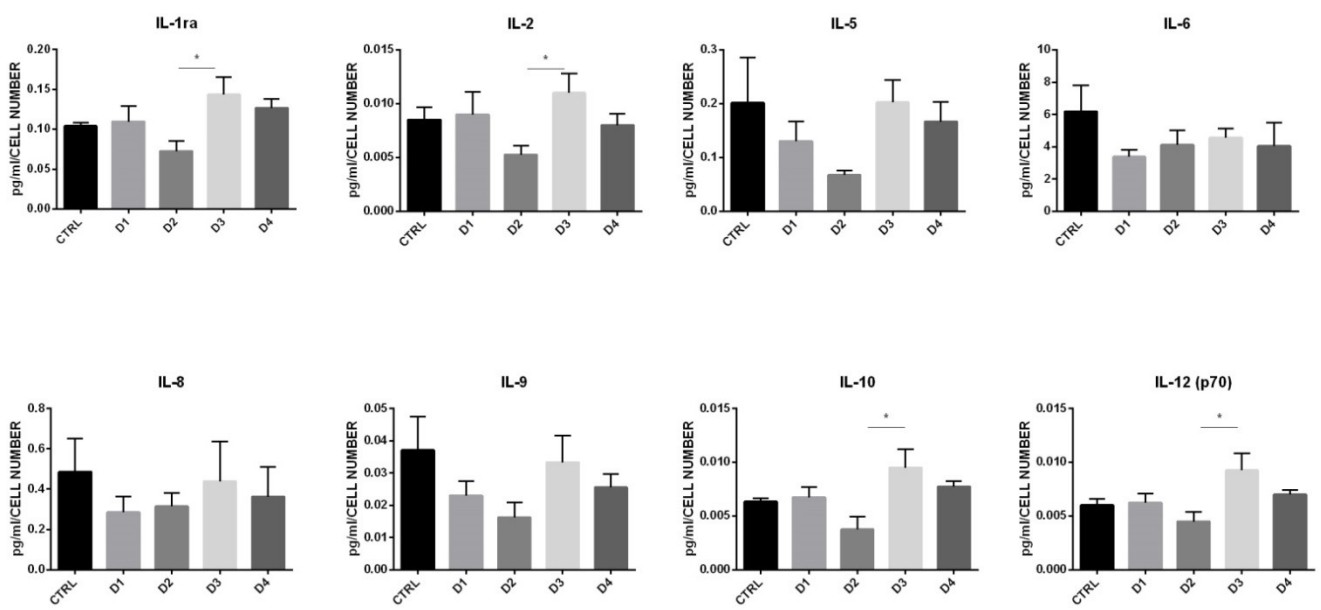

**Figure 5.** *Cont*.

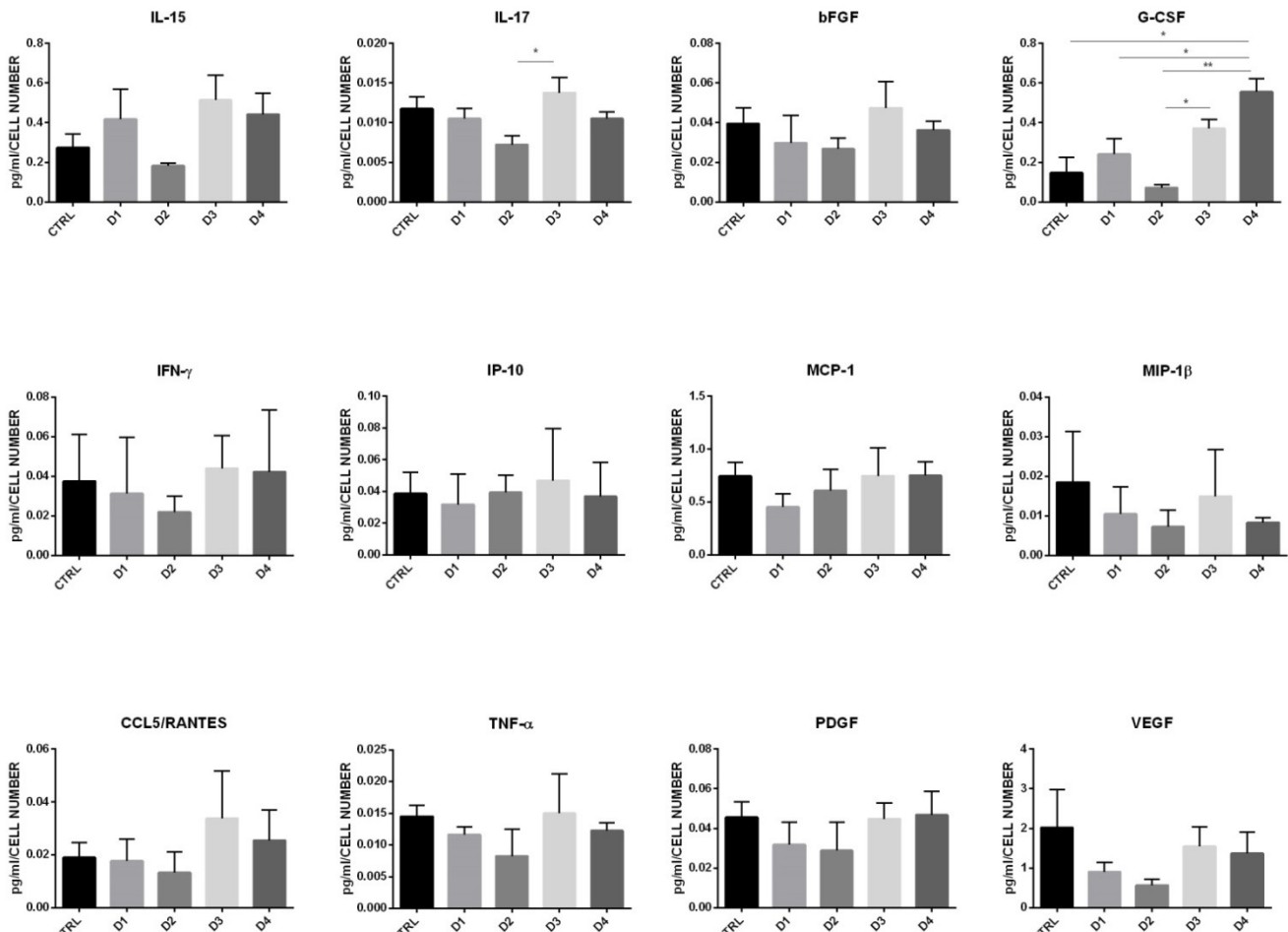

**Figure 5.** BFP-MSC-released cytokines, chemokines and growth factors. BFP-MSCs were plated in 12-well culture plates without disks (CTRL), on D1, D2, D3 or D4 for 72 h. Supernatants were collected and tested by using the Bio-Plex multiplex cytokine assay kit. * Denote statistical differences (* $p < 0.05$; ** $p < 0.01$).

## 4. Discussion

Many studies have reported that the bone–implant interface can influence peri-implant bone healing and osseointegration [7–10]. Different treatment techniques of Ti surfaces were suggested to improve the peri-implant bone healing after implant placement; their aim is to increase the contact area between the implant surface and living bone and to promote the adhesion of biological molecules capable to stimulate tissue regeneration [6,8,15]. Manufacturers have introduced different treatments, both chemical (acid-etching) and mechanical (grit-blasting) or a combination thereof for improving bone contact/anchorage compared to a smooth dental implant surface [15]. Anodization, physical vapor deposition, calcium phosphate coating, and hydroxyapatite coating are other surface-treatment methods commonly used [25]. The proposed engineering processes influenced the implant surface roughness and the chemical composition. The acid-etching treatment seemed to develop the coarsest implant surface with a spiky and sharp-cornered morphology, while a fluctuating morphology occurred with sandblasting. Indeed, implant surface obtained with the acid etching process did not present any residue from the performed chemical treatment; by contrast, in sandblasted surfaces there could be a potential risk of surface contamination with the presence of blasting particle remnants [15,26]. Their biologic/clinical value is not clear; the inhibition of bone mineralization, the activation of osteoclast-like cells and the improvement of the bone erosion were related to their presence. The reported

results evidenced how Kontact surface (grit-blasted) showed a lower adhesivity compared to Kontact S (acid-etched) and contact N (blasted/etched) surfaces. This could be related to a greater degree of contamination by alumina blasting particle remnants of the Kontact surface. As the chemical composition and contamination can be considered aspects that significantly influence the biocompatibility of the fixture, its roughness and wettability guide the differentiation of mesenchymal stem cells and influence the type, quantity and conformation of the absorbed protein layer [11,13,27–30]. Based on the literature, mesenchymal stem cells from adipose tissue were utilized as a substitute to human bone marrow MSCs for assessing the osseointegration in Ti implants [17,19,29–32]. This research evidenced how BFP-MSCs cultured on different Ti implant surfaces, of different chemical compositions and surface treatments, presented similar cellular adhesion. However, Kontact S (chemically etched surface), and Kontact N (blasted/etched surface), which presented higher values of roughness parameters, showed a statistically significant greater adhesion capability than Kontact (grit blasted surface). Until now, the relationship between the surface roughness and its ability to promote the osseointegration process has not been clear. Many authors evidenced an ability to increase the contact area between the implant surface and living bone and to promote the cytokines release by immune cells [33–35]. MSCs are characterized by high immunomodulatory functions, exerted through the cytokines and chemokines release, which in turn modulate the local microenvironment, including the immune cell properties [36,37]. Cytokines and chemokines play a crucial role in bone wound healing, acting as messengers between inflammatory cells, keratinocytes, fibroblasts, endothelial cells and stem cells [38]. These soluble mediators control the inflammation process, the motility, proliferation and matrix deposition of all the cells involved in bone regeneration [39]. For instance, IL-1ra and IL-10 have well known anti-inflammatory functions [40,41]; IL1ra, physiologically secreted by the anabolic macrophage phenotype M2 (with anti-inflammatory and pro-regenerative phenotype) and by the recruited stromal stem cell contributes to the formation of a fibro-vascular granulation tissue ultimately resulting in local bone formation [42]. IL-2 is involved into both the induction and the termination of inflammatory immune responses and play a key role in favoring wound healing [43,44]. IL-12 is a pro-inflammatory factor that amplifies downstream inflammatory signals and impairs MSC-mediated bone regeneration [45]. Conversely, IL-17, released by $\gamma\delta$ T cells after bone damage, enhances osteogenic differentiation of resident MSCs [46]. In addition, basic FGF has a crucial role for the maintenance of stem cell features, as well as for bone regeneration [47]. In bone regeneration there is not a dominant cytokine, but modulation drive matrix deposition and wound healing. In vivo, in peri-implant sites their release can be influenced by certain factors: age, diabetes, smoking habits, oral hygiene, preoperative and postoperative antibiotics administration, some surgical aspects (flap technique, drilling protocols, insertion torque) [48]. The relationship between the surface parameters (roughness, wettability, chemical composition) and their ability to influence the cytokines release aimed to promote the osseointegration process is unclear. Our results showed that BFP-MSCs release detectable levels of a large variety of cytokines, chemokines and growth factors and that the implant surfaces do not interfere with their secretory functions. For example, no differences were found in the release of IL-5, involved in eosinophil functions [49], of IL-9, implicated in inflammatory diseases and in guarding immune tolerance [50], and of IL-15, required for osteoblast function and bone mineralization [51]. Nevertheless, in presence of Kontact S surface (D2-chemically etched surface), cells released a lower amount of IL-1ra, IL-2, IL-10, IL-12, IL-17, G-CSF, and basic FGF compared to cells plated on D3 (sandblasted surface). Interestingly, in presence of D4 (blasted/etched surface) BFP-MSCs released significantly higher concentrations of G-CSF, compared to cells plated without disk and to cells plated on D1 and D2. G-CSF is a glycoprotein that is used therapeutically for its ability to mobilize hematopoietic, mesenchymal and vascular stem cells in the systemic circulation. It has a proliferative effect on osteoprogenitors and enhances bone regeneration via revascularization and osteogenesis [52,53]. As result, the increased levels of this growth factor may have a crucial role for the improvement of

peri-implant bone healing. The Kontact N surface, for its roughness, adhesivity capacity and grow factors stimulation, seemed to assure the best outcomes in comparison to the others examined implant surfaces.

In future, research should clarify the correlation between the biological response and the implant surface characteristics; it would be the ideal situation from an industrial standpoint and the development of a clear and simple ISO standard.

The strength of this study is represented by the use of primary mesenchymal stem cells obtained from the Bichat fat pad and by the evaluation of a large number of cytokines, chemokines and growth factors. Indeed, beside cell viability, the MSC secretory pattern is an important parameter to consider in order to achieve tissue regeneration. Thus, the determination of an MSC secretory phenotype may represent a novel and relevant functional assay. The main limitation is represented by the limiting number of not reusable disks, which did not allow measuring cell proliferation as a function of time and by the lack of a straight methodology for direct visualization and counting of cells. However, the use of crystal violet staining and of an automated cell counter are largely used methods to visualize as well as to evaluate cell number and viability.

### 5. Conclusions

- Overall, BFP-MSCs seeded in the presence of Ti disks showed a proliferative rate and viability comparable to cells plated without disks.
- When plated on Kontact S (chemically etched surface) and Kontact N (blasted/etched surface) the number of living cells counted on disks was significantly higher compared to that of cells remaining under the disks.
- Once detached from Ti disks, BFP-MSCs retained their morphology and viability.
- Culturing on Ti disks preserved secretion of multiple cytokines and chemokines by BFP-MSCs. This is of relevance since other studies have shown that implant surfaces may enhance the release of some inflammatory molecules by immune cells.
- In the presence of Kontact N (blasted/etched surface) BFP-MSCs released significantly higher concentrations of G-CSF, compared to cells plated without disks and to cells plated on a smooth Ti surface and Kontact S surface.

Further in vitro and in vivo studies are necessary to define the implant surface parameters able to assure their optimal release for a real improvement of dental implant osseointegration. However, these novel findings, bridging together medicine, stem cell biology, and material sciences, provide new knowledge regarding the interaction between Ti implants and BFP-MSCs. This approach will contribute to paving the way for novel therapeutic strategies and oral surgery applications.

**Author Contributions:** Conceptualization, V.D.; methodology, J.C.S.; investigation, A.P. and D.L.; data curation, P.F.; writing—original draft preparation, G.S. and D.A.; writing—review and editing, G.M. All authors have read and agreed to the published version of the manuscript.

**Funding:** This research received no external funding.

**Institutional Review Board Statement:** Not applicable.

**Conflicts of Interest:** The authors declare no conflict of interest.

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
