# Peer review of "Effect of Different Titanium Dental Implant Surfaces on Human Adipose Mesenchymal Stem Cell Behavior. An In Vitro Comparative Study"

_applsci, doi:10.3390/app11146353_

Round 1

Reviewer 1 Report

If the authors are not able to visualize cells growing over the disk, how can they say that cells are better attached in one disk than in another one?

Do the authors think that a trypan blue assay is the best way to evaluate the biocompatibility of these materials?

Why didn’t the authors evaluate the cytotoxicity of cells growing over the Ti? The indirect measurement of the re-plated cells might not be a real indicator of this state.

Why the D3 sample showed the worst biocompatibility results? Did the Alumina present in the process might have interfered?

The surface roughness of the samples is needed and should be discuss related to the adhesion and proliferation of the cells.

Contact angle measurements are needed and should be discussed.

What is the purpose of figure 4? What can be said about the fact that cells are able to proliferate after a re-seed?

In general terms, which surface treatment presents the best outcome and why?

I would advise improving the graphics, also images in Figure 3 are of low quality and scale bars should be more visible. 

Author Response

Comments and Suggestions for Authors

Referee 1

  1. If the authors are not able to visualize cells growing over the disk, how can they say that cells are better attached in one disk than in another one? Do the authors think that a trypan blue assay is the best way to evaluate the biocompatibility of these materials? Why didn’t the authors evaluate the cytotoxicity of cells growing over the Ti? The indirect measurement of the re-plated cells might not be a real indicator of this state.

Thank you for your comments. We have added more details on method used for evaluating cell number and viability, both in Materials and methods (page 3, 133-142 lines)  and in results (page 4, 172-178 lines). We have inserted also a new reference (ref 23) regarding the protocol and the use of trypan blu staining. Moreover, we have added the reference 22 relative to the use of this staining on cells plated on Ti surfaces. In addition, we have clarified that, by using the automated cell counter, we were able to visualize stained and not stained cells (page 4, 172-178). Finally, a comment on cytotoxicity was inserted (page 5, 10-192 lines)

  1. Why the D3 sample showed the worst biocompatibility results? Did the Alumina present in the process might have interfered?

thank you for your suggestionThe lower adhesivity of the Kontact surface was related to a greater degree of contamination by alumina. (Page 9, 253-256 lines)

  1. The surface roughness of the samples is needed and should be discuss related to the adhesion and proliferation of the cells.

Thank you for your suggestion.  Roughness measurement of the dental implants surfaces were reported in Tab 1 (page 3, lines 101-109)

  1. Contact angle measurements are needed and should be discussed.

Thank you for your suggestion.  Ti disks were completely immersed in culture medium for this reason we have not considered this parameter

  1. What is the purpose of figure 4? What can be said about the fact that cells are able to proliferate after a re-seed?

Thank you for your suggestion.  The purpose of the experiment in fig 4 was to provide evidence that MSCs were still alive and able to proliferate also upon the contact and detachment from Ti disks. We have better clarified this point at (page 5, 199-202 lines)

  1. In general terms, which surface treatment presents the best outcome and why?

Thank you for your suggestion.  This opinion was reported in page10, 309-311 lines

  1. I would advise improving the graphics, also images in Figure 3 are of low quality and scale bars should be more visible. 

Thank you for your suggestion.  The figure has been modified as layout and enlarged

Referee 2

Reviewed article is interesting and write at proper scientific level. Presentation method is mostly good and in accordance with generally accepted standards in that area. Figures, tables as well as terminology are clear and precise. Below are listed some substantive remarks that should be taken into consideration by the Authors to improve reviewed text:

  1. at the end of the introduction should be clearly and concise given the research gap to create the appropriate lead up for the motivation of the work.

Thank you for your suggestion. It was reported in page 2, line 74-75

  1. the novelty of given approach should be emphasized in introduction,

Thank you for your suggestion.  Done, accordingly (page 2, line 78-80)

  1. I suggest to provide more precise information about used experimental and measurement positions,

Thank you for your suggestion. We have added more details in materials and methods section. (page 3, lines 133-142; pag 4, lines 172-178)

  1. I suggest also to give wider description of potential use of presented findings in scientific research as well as in industrial practice

Thank you for your suggestion. The industrial finality and the aim of the research were reported in pag10, lines 312-314

  1. in the discussion section should be provide more references to already known results from literature,

As suggested, in discussion section refs 25, 26, 37 and 38 have been added. Moreover refs 22 and 23 have been added in Materials and methods section

  1. the strengths and limitations of the obtained results and applied methods should be clearly described,

Thank you for your suggestion.  Done, accordingly (page 9, 315-324 lines)

  1. I suggest to provide the main conclusions as numbered sentences and refer to specific values (results of analysis) as well as basic phenomena that cause described results,

Thank you for your suggestion.  Done, accordingly (page 10/11, 328-341 lines)

  1. the conclusion should be improved in term of the new knowledge gained during analysis, which should be concise with the journal scope.

Thank you for your suggestion.  Done, accordingly (page 9-10, 328-341 lines)

Reviewer 2 Report

Reviewed article is interesting and write at proper scientific level. Presentation method is mostly good and in accordance with generally accepted standards in that area. Figures, tables as well as terminology are clear and precise. Below are listed some substantive remarks that should be taken into consideration by the Authors to improve reviewed text:

  • at the end of the introduction should be clearly and concise given the research gap to create the appropriate lead up for the motivation of the work,
  • the novelty of given approach should be emphasized in introduction,
  • I suggest to provide more precise information about used experimental and measurement positions,
  • I suggest also to give wider description of potential use of presented findings in scientific research as well as in industrial practice,
  • in the discussion section should be provide more references to already known results from literature,
  • the strengths and limitations of the obtained results and applied methods should be clearly described,
  • I suggest to provide the main conclusions as numbered sentences and refer to specific values (results of analysis) as well as basic phenomena that cause described results,
  • the conclusion should be improved in term of the new knowledge gained during analysis, which should be concise with the journal scope.

Author Response

(The authors gave the same response as above.)

Round 2

Reviewer 2 Report

All my comments were taken into consideration by the Authors.